# High prevalence of caesarean birth among mothers delivered at health facilities in Bahir Dar city, Amhara region, Ethiopia. A comparative study

Meseret Bantigegn Melesse[1]*, Alehegn Bishaw Geremew[2], Solomon Mekonnen Abebe[3]

**1** Zonal Health Department, Bahir Dar city, Ethiopia, **2** Department of Reproductive Health, Institute of Public Health, College of Medicine and Health Sciences, University of Gondar, Gondar, Ethiopia, **3** Department of Human Nutrition, Institute of Public Health, College of Medicine and Health Sciences, University of Gondar, Gondar, Ethiopia

* alexbishaw@gmail.com

## Abstract

### Objective

The study aimed to assess and compare the prevalence of caesarean birth and associated factors among women gave birth at public and private health facilities in Bahir Dar city, Amhara region, Ethiopia.

### Methods

An institution-based comparative cross-sectional study design was conducted from March1-April 15, 2019 at health facility provide emergency obstetrics service in Bahir Dar city. Study participants 724(362 for each public and private facility) were recruited using a systematic random sampling technique. Structured interview administered questionnaires and chart review checklist were used to collect data. The data were entered into Epi info version 7.2 and analyzed using SPSS version 23.0 software. A binary logistic regression model was fitted and an adjusted odds ration with 95% CI was used to determine the presence and strength of association between independent variables and cesarean birth.

### Results

The response rate was 98.3% and 97.2% for public and private health facilities respectively. The prevalence of caesarean birth in private health facilities was 198 (56.3%) (95%CI: 50.9, 61.4) and in public health facilities was 98 (27.5%) (95%CI: 22.8, 32.2). Overall prevalence of caesarean birth was 296 (41.8%) (95%CI: 38.4, 45.5). Breech presentation (AOR = 3.64; 95%CI:1.49, 8.89), urban residence (AOR = 6.54; 95%CI:2.59, 16.48) and being referred (AOR = 2.44; 95%CI:1.46, 4.08) were variables significantly associated with caesarean birth among public facilities whereas age between 15–24 (AOR = 0.20, 95% CI: 0.07, 0.52), government employe (AOR = 2.28; 95%CI: 1.39,3.75), self-employed (AOR = 3.73; 95% CI:1.15,8.59), para one (AOR = 6.79; 95%CI:2.02, 22.79), para two (AOR = 3.88; 95%

**Funding:** The source of funding for this study was Amhara regional health bureau. The funder had no role in study design, data collection and analysis, decision to publish, or preparation of the manuscript.

**Competing interests:** The authors have declared that no competing interests exist.

**Abbreviations:** AOR, Adjusted Odd Ratio; APH, Antepartum Hemorrhage; CPD, Cephalic Pelvic Disproportion; COR, Crude Odd Ratio; CS, Cesarean Section; EDHS, Ethiopia Demographic and Health Survey; EmONC, Emergency Obstetric and Newborn Care; NRFHRP, Non Reassuring Fetal Heart Rate Pattern; PPH, Postpartum Hemorrhage; PROM, Premature Rupture of the Amniotic Fluid Membrane; SPSS, Statistical Package for Social Science; VBAC, Vaginal Delivery After Cesarean Section; VD, Vaginal Delivery; WHO, World Health Organization.

CI:1.15,13.08), and wealth index being highest level of wealth asset AOR = 5.39; 95% CI:1.08, 26.8) in private health facility associated with caesarean birth.

## Conclusions

We concluded that there is high prevalence of caesarean birth both in private and public facility. There is a statistically significant difference in the prevalence of caesarean birth in public and private health facilities.

## Introduction

Caesarean section (CS) is an operative technique by which a fetus is delivered through an abdominal and uterine incision of the mother[1]. When adequately indicated caesarean section is one of the life-saving procedures that attributed to the decrease of the maternal and neonatal mortality and morbidity rates globally [2].Cesarean section was first major operation for high-risk pregnancy. But the capability to perform safe CS has been one of the major advances in obstetrics in the 20th century and contributed to the more frequent use of the procedure worldwide[3]. The safety of the operation has improved with time, largely due to improved surgical and anesthetic techniques[4].

World Health Organization (WHO) suggested that the rates of the caesarean section should not exceed 15% because has no additional benefit for the newborns or the mothers. On the other hand, a rate of less than 5% would reflect the difficulty in access to adequate treatment [5]. However, WHO released a statement indicating that at the population level, rates higher than 10% were not associated with reductions in maternal and newborn mortality rates in developing countries[6].

Despite, the WHO recommendation many works of literature have shown that CS rates are rising in developing and low-income countries, just as in their developed country counterpart [7]. Sub-Saharan Africa still has the lowest rates of caesarean birth, with many countries having national CS rates below 5%[8]. There is an inequitable distribution of caesarean birth rate, even within the poor countries; with urban resident women having better access and more CS deliveries than their rural neighbors in Ethiopia[9, 10].

The reasons for the rise in the rate of caesarean section birth include in part an increase in the facility-based delivery and access to health care [11]. The national prevalence of caesarean birth in Ethiopia is far below the WHO optimum range, 5–15%[12]. The caesarean birth rate figures across the sub-national regions are variable, ranging from <1% - 25%[9].

Ethiopia is among the countries having good progress in reducing maternal mortality with access to obstetric care including caesarean sections delivery [10].

Cesarean sections are comprehensive obstetrics care service which, prevent both maternal and neonatal morbidity and mortality. However, there are short and long-term risks and high cost associated with caesarean birth, and there are no health benefits of CS when the procedure is performed without a medical indication, and there is evidence that maternal death and disability is higher after CS than vaginal birth[13]. Studies show the caesarean section (CS) prevalence rate has been an alarming increase worldwide each year[14].

The national prevalence of cesarean section birth among very low-income countries like Ethiopia requires specific attention, considering that access to caesarean birth is still insufficient. However, caesarean birth seems to rise inappropriately in private facilities and some urban settings[15]. Context where caesarean birth rate less than %% and greater than 15% are

unwanted and it is important to understand the underlining causes to put in place interventions to prevent maternal morbidity and mortality [16]. CS birth is increasing in Ethiopia, which is indicative of access to obstetric care service in the country[17].

WHO published the first new global guidelines on non-clinical interventions, specifically designed to reduce unnecessary CSs birth[18]. However, a little has been known about factors associated with increase CS birth and there is limited information concerning the prevalence of CS birth in public and private health facility in Amhara region. Therefore, this study aimed to assess and compare caesarean section birth and its associated factors among women gave birth in public and private health facilities in Bahir Dar city.

## Methods and materials

### Study area

The study was conducted at health facilities in Bahir Dar city. Bahir Dar is the capital city of the Amhara National Regional State in the Federal Democratic Republic of Ethiopia. According to the Amhara Bureau of Finance and Economic Development (BOFED), the population of Bahir Dar city was estimated to be 339,683. Among these, 156,376(46%) of them are females. The city has one specialized, one referral and one primary government hospitals (Tibebe Giwon, Felege Hiwot,and Adiss Alem respectively), 11 health centers (including one private health center), 10 health posts and one family guidance association clinic, 4 private general hospital, and 35 medium private clinics. Among all health facilities 4 public facilities named Tibebe Giwon, Felege Hiwot, Adiss Alem, Bahir Dar health center, and 5 private health facilities named GAMBY hospital, Mari stop, Addinas General hospital, and Dr. Amiro MCH specialty clinic were provide Emergency Obstetric and Newborn Care service during the study period. According to the Bahir Dar city zone health department 2010 E.C report, there were 15,208 annual deliveries and among this 4,160 had CS birth [19].

### Study design and period

An institutional-based comparative cross-sectional study was conducted from March 1 to April 15, 2019.

### Population

All women who gave birth in public and private health facilities providing caesarean birth services in Bahir Dar city were the source of population. Women who gave birth in selected public and private health facilities in Bahir Dar city during the study period was the study population

### Sample size determination

The sample size was estimated using double proportion formula; considering caesarean birth proportion (public facility 34% and private facility 47%) from previous study in Addis Ababa [17]. The formula and calculation as follow:

N (in each group) = $(p_1q_1 + p_2q_2) (f(\alpha,\beta)) / ((p_1 - p_2)^2$

Where n = sample size for each group

P1 = the proportion of CS at Private health facilities (0.47).

P2 = the proportion of CS at Public health facilities (0.34).

F ($\alpha,\beta$) = 7.84, when the power = 80% and the level of significance = 5%

q1 = (1-p1) = 1–0.47 = 0.53

q2 = (1-p2) = 1–0.34 = 0.66

n1 = n2 = $(0.84+1.96)^2 ((.47 \times .53) + (.34 \times .66)) / (.47 - .34)^2 = 219$

Total sample size for both groups = 438

Multiplied by design effect of 1.5 I.e. 438 \*1.5 = 657

The sample size for factors associated with caesarean birth was also calculated using Epi info version 7.2.2 and found to be less than the sample size for the proportion of caesarean birth. Therefore, by adding 10% none response rate, the final estimated total sample size for this study was 724 (362 and 362 study subjects for the public and private health facility)

## Sampling procedure

A multistage systematic random sampling procedure was used. From a total of 9 health facilities (4 public and 5 private) which provide a comprehensive obstetric and newborn care in Bahir Dar city, 3 public and 4 private health facilities were selected using simple random sampling. The required sample size from each group was proportionally allocated using stratified sampling for selected health facilities in each group, based on the previous year's week's average number of client flow. The six-week average client flow of the selected health facilities, Flege Hiwot referral Hospital, Addis Alem hospital, Bahir Dar health center, GAMBY hospital, Mari Stop, Addinas General hospital, and Dr. Amiro MCH specialty clinic were 521, 254, 363, 72, 393, 173 and 124 respectively. The proportional allocation was done for each facility in each stratum. Systematic random sampling was used to select each study subject. The first case was randomly selected after calculating the interval for both public and private facility and then every 3rd case for public and every 2nd case for private health facilities were selected from the delivery record till the required sample size was achieved for each facility.

## Operational definitions

Medical or obstetrical indication is considered at least one of the following is occurred:- obstructed labor or cephalo-pelvic disproportion or antepartum hemorrhage or previous cesarean section scar or mal-presentation or preeclampsia/ eclampsia syndrome or failure to progress or failed induction or suspected uterine rupture or cord prolapsed or non-reassuring fetal heart rate pattern and post-term [2]. Cesarean delivery on maternal request is defined as a primary cesarean birth done on request from the mother in the absence of any medical or obstetric indication[6].

## Data collection tool and procedure

The questionnaire and checklist were adapted through reviewing of different works of literature and previous similar studies [17, 20–24]. The questionnaire was initially prepared in English, then translated to Amharic, and then translated back into English to check for consistency. A structured questionnaire was used to collect the data through a face to face interview and checklist for reviewing client charts. The Amharic version of the questionnaire was used for data collection. The main variables included in the questionnaire and checklist for assessment were: socio-demographic characteristics of the respondents, previous and current pregnancy history, indication of caesarean birth and fetal conduction (S1 File).

The data was collected by trained seven diploma midwives working in obstetric wards of other health facilities. The participants were interviewed after 2 hours of vaginal birth and 6 hours of caesarean birth until women were discharged from the health facility considering stable to communicate. The checklist was filled after the delivery summary was written by the clinician. The data collection process was supervised by two BSc holder senior staff working in the obstetric department. The questionnaires, and checklist filled completely were collected daily after checking the completeness, and consistency of the data.

## Data quality assurance

To maintain quality of data the data collectors and supervisors were trained for two days on the objective of the study, the content of the questionnaire, how to fill the questionnaire, respondent rights, informed consent, and technique of interview and how to keep confidentiality and privacy of the study subjects. Before one week of the actual data collection period, the data collection tools were pretested on 36 individuals in Tibebe Giwon and Dream care Hospitals then possible adjustment or modification like skipping interval, order of questions list was made. The principal investigators & supervisors gave feedback and correction daily for the data collectors. The data was cleaned, coded and entered to Epi info version 7.2.2.

## Data processing and analysis

Each completed questionnaire was coded on a pre-arranged coding sheet by the principal investigator to minimize errors. Data were entered into a computer using Epi info version 7.2.2, and then exported to statistical package for social science (SPSS) version 23.0 for further cleaning and analysis. The data were cleaned by sorting, cross tabulation, and after the data were cleaned frequencies and percentages has generated. Mean and standard deviation measure of summary were used after checking the nature of the data. According to the variables the findings presented by text, tables and graphs

Initially, bivariable logistic regression analysis was performed between the dependent variable and each of the independent variables. Then all variables with a p-value<0.05 from bivariable logistic regression analysis were fitted into the multivariable logistic regression model to control possible confounders and backward regression analysis were done Adjusted odds ratio (AOR) with 95% confidence interval (CI) was used to measure the strength and significance of the association. A P-value <0.05, also has indicated the presence of a statistically significant association between caesarean birth and independent variables. Chi-square tests was used to determine statistical difference of caesarean birth between private and public facility delivery.

## Ethical approval and consent to participate

Ethical clearance was obtained from the Institute of Public Health, College of Medicine and Health Sciences on the behalf of University of Gondar ethical review Board (IRB) number IPH/180/06/2022. A written support letter was obtained from Amhara regional health bureau, Bahir Dar city administrative zone, health department, and each respective health facility office of administration. After the purpose and objective of the study have been informed, a verbal consent was obtained from each study participant. Participants were also informed that their participation was voluntarily and they can stop or leave from the participation at any time if they are not comfortable. The data collection tools were anonymous and keeping participant's privacy during the interview by interviewing them alone to keep the confidentiality of any information provided by participants.

## Results

### Socio-demographic characteristics of the respondents

A total of, 708 women has participated in this study with the overall response rate of 97.8%. The response rate of public and private facility was 98.3% and 97.2% respectively. The mean age of the respondents was 27.31 with SD ± 5.01 years for public and 29.27 with SD± 4.53 years for private health facility. Concerning participants' residency, 82.6% in public health facilities and all private health facilities respondents were urban resident. Out of the

**Table 1. Socio-demographic characteristics of women who give birth in the selected public and private health facilities in Bahir Dar city, Amhara Regional State, Ethiopia, 2019 (n = 708).**

| Variables | Category | Public health facilities (N = 356) | Private health facilities(N = 352) | Total (N = 708) |
|---|---|---|---|---|
| | | Frequency (%) | Frequency (%) | Frequency (%) |
| Age in years | 15–19 | 11(3.1) | 2(.6) | 13(1.8) |
| | 20–24 | 96(27.0) | 55(15.6) | 151(21.3) |
| | 25–29 | 137(38.5) | 140(39.8) | 277(39.1) |
| | 30–34 | 71(19.9) | 98(27.8) | 169(23.9) |
| | 35–39 | 41(11.5) | 57(16.2) | 98(13.8) |
| Residency | Rural | 62(17.4) | - | 62(8.8) |
| | Urban | 294(82.6) | 352(100) | 646(91.2) |
| Marital status | Single | 6(1.7) | 1(0.3) | 7(1.0) |
| | Married | 344(96.6) | 350(99.4) | 694(98.8) |
| | Divorced | 3(0.8) | 1(0.3) | 4(0.6) |
| | Widowed | 3(0.8) | - | 3(0.4) |
| Women Educational status | No formal education | 107(29.6) | 3(0.9) | 110(15.5) |
| | Primary school (1–8) | 89(24.6) | 39(11.1) | 128(18.1) |
| | Secondary9-12) | 92(25.4) | 95(26.9) | 187(26.4) |
| | Collage and above | 68(18.8) | 215(61.1) | 283(40.0) |
| Women Occupation | Housewife | 189(53.1) | 154(43.8) | 343(48.4) |
| | Government employee | 44(12.4) | 157(44.6) | 201(28.4) |
| | Private employee | 27(7.6) | 26(7.4) | 53(7.5) |
| | Farmer | 41(11.6) | - | 41(5.8) |
| | Merchant | 37(10.4) | 14(4.0) | 51(7.2) |
| | Daily labour | 10(2.8) | - | 10(1.4) |
| | Student | 8(2.2) | 1(0.3) | 9(1.3) |
| Spouse educational status | No formal education | 99(28.6) | 3(0.9) | 102(14.6) |
| | Primary school (1–8) | 53(15.3) | 12(3.4) | 65(9.5) |
| | Secondary (9–12) | 83(24.0) | 59(16.8) | 142(20.3 |
| | College and above | 111(32.1) | 277(78.7) | 388(55.70 |
| Spouse occupation | Government employee | 97(28.2) | 182(51.7) | 279(40.1) |
| | Self-employee | 85(24.7) | 106(30.1) | 191(27.5) |
| | Farmer | 59(17.2) | - | 59(8.5) |
| | Merchant | 88(25.6) | 57(16.2) | 145(20.9) |
| | Others* | 15(4.2) | 6(1.7) | 21(3.0) |
| Wealth index | Lowest | 76(21.3) | 9(2.6) | 85(12.0) |
| | Second | 68(19.1) | 111(31.5) | 179(25.3) |
| | Medium | 66(18.5) | 67(19.0) | 133(18.8) |
| | Fourth | 103(28.9) | 103(29.3) | 206(29.1) |
| | Highest | 43(12.1) | 62(17.6) | 105(14.8) |

*Daily labour, none governmental organization

participants, 68 (18.8%) of public and 215(65.1%) of private health facility respondents had educational status were college diploma and above (**Table 1**).

## Obstetric related factors

Regarding the obstetrics factors of the participants, 156 (43.8%) of them were para one, and 65 (18.3%) of women was para four and above in public facilities whereas in private health

**Table 2. Obstetrics factors of women who delivered in the selected public and private health facility provide cesarean birth service in Bahir Dar city, Amhara regional state, Ethiopia, 2019 (n = 708).**

| Variables | Category | Public health facility (n = 356) | Private health facilities(n = 352) | Total<br>N = (708) |
|---|---|---|---|---|
| | | Frequency (%) | Frequency (%) | Frequency (%) |
| Parity | Para 1 | 156(43.8) | 187(53.1) | 343(48.4) |
| | Para 2 | 85(23.9) | 101(28.7) | 186(26.3) |
| | Para 3 | 50(14.0) | 48(13.6) | 98(13.8) |
| | Para ≥4 | 65(18.3) | 16(4.5) | 81(11.4) |
| Gestational age | Pre term | 31(8.7) | 3(85.2) | 34(4.8) |
| | Term | 321(90) | 343(97.4) | 664(93.7) |
| | Post-term | 4(1.1) | 6(1.7) | 10(1.4) |
| Onset of labor | Spontaneously | 296(83.1) | 264(75.0) | 560(79.0) |
| | Induced | 60(16.4) | 88(25.0) | 148(20.9) |
| ANC visit | Yes | 345(96.9) | 352(100) | 697(98.4) |
| | No | 11(3.1) | - | 11(1.6) |
| No of ANC | One | 7(2.0) | - | 7(1.0) |
| | Two | 28(8.0) | - | 28(4.0) |
| | Three | 56(16.2) | 9(2.6) | 65(9.3) |
| | Four and above | 254(73.6) | 343(97.4) | 597(85.7) |
| Abortion history | Yes | 16(7.8) | 13(7.7) | 29(7.8) |
| | No | 188(92.2) | 155(92.3) | 343(92.2) |
| Previous history infertility | Yes | 6(1.7) | - | 6(0.8) |
| | No | 350(98.3) | 352(100) | 702(99.2 |

facilities 65(18.3%) was para one and 13(3.7%) of women was para four. Among women from public 345(96.9%) and private health facilities 352(100%) had at least one antenatal care (ANC) follow up during pregnancy of current delivery. Among women who had ANC follow up 254(73.6%) in public and 343 (97.4%) in private health facilities participants had four ANC visits. The majority of mothers 321(90%) gestational age at the time of delivery in public and 343(97.4%) in private health facility were term pregnancy (**Table 2**).

## Prevalence of caesarean birth

The prevalence of caesarean birth in public health facilities was 98 (27.5%) (95% CI: 22.8, 32.2) and in private health facilities was 198(56.3%) (95%CI: 50.9, 61.4). The difference in the prevalence of cesarean birth among public and private health facilities was significantly different (P<0.001). Caesarean birth was much more in private health facilities. The overall prevalence of caesarean birth was 296 (41.8%) (95% CI: 38.4, 45.5). Out of caesarean birth performed in public health facilities, 89(90.8%) was an emergency and the rest 9(9.2%) was elective, as compared to 125(63.1%) emergency and 73(36.9%) elective caesarean birth performed in private health facilities (**Fig 1**).

## Indication of caesarean section birth

Out of women had given birth by caesarean section in public health facility: none reassuring fetal heart rate (NRFHR) 24(24.5%). breech presentation 23(23.5%), obstructed labor 15 (15.3%), previous CS12(12.2%), CPD 8(8.2%) preeclampsia/ eclampsia 5(5.1%), post term pregnancy and APH each 4(4.1%) and twin pregnancy 1% were the indication. In private health facilities the indications were previous cesarean section scar 51(25.8%), NRFHR 50

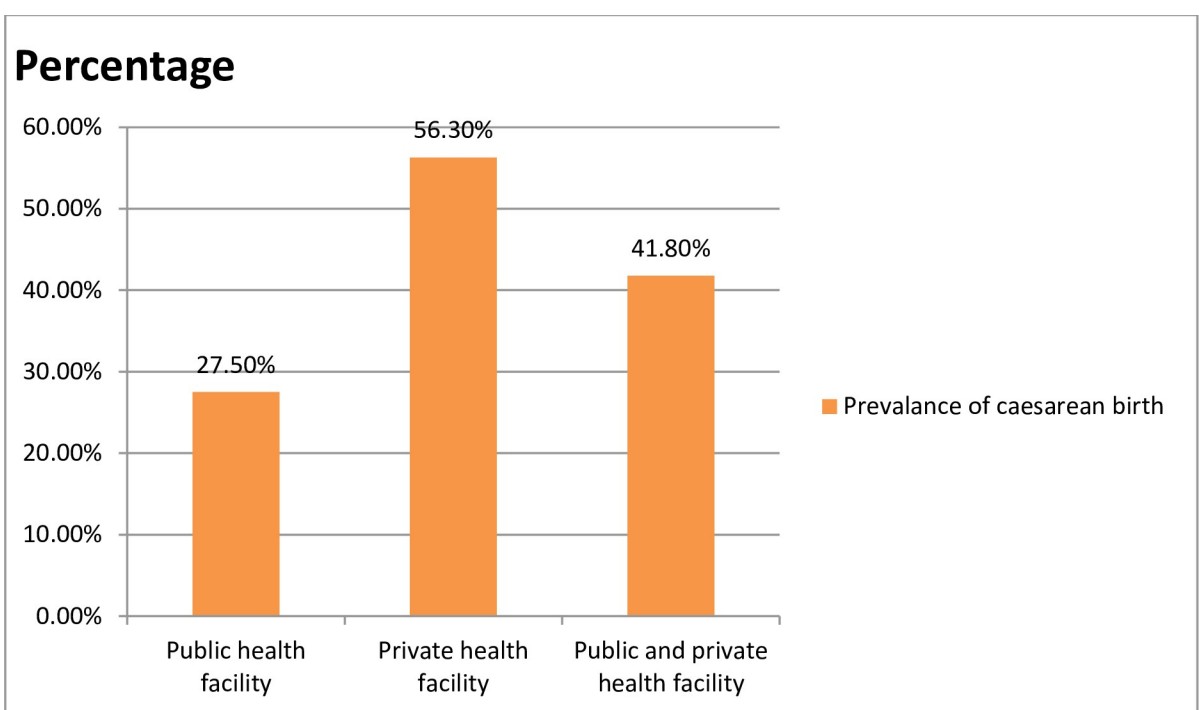

**Fig 1. Prevalence of caesarean birth among women who delivered in the selected public and private health facilities in Bahir Dar city, Amhara regional state, Ethiopia.**

(25.3%), cephalo-pelvic disproportion 40(20.2%), breech presentation 19(9.6%), obstructed labor 12(6%), post term pregnancy 6(3%), preeclampsia/ eclampsia 4 (2%) and 16(8.1%) were due to maternal request (**Fig 2**).

## Factors associated with caesarean section birth

Findings from the multivariable analysis of public health facility, variables statistically significant were fetal presentation, breech presentation were 3.64 times more likely to have CS birth than cephalic presentation (AOR = 3.64 (95%CI 1.49, 8.89)), urban residence were 6.54 times more likely to have CS birth than rural (AOR = 6.54(95%CI 2.59, 16.48)) and referral status: being referred 2.44 times more likely to have CS birth than not referred women (AOR = 2.44 (95%CI 1.46, 4.08)).

In the multivariable analysis of private health facilities variables remained statistically associated with caesarean birth were: age; women aged 15–24 years was 80% (AOR = 0.20 (95% CI; 0.07, 0.52)) less likely to have CS birth as compared to women aged 35 years and above. Women occupation; being governmental employees 2.28 times more likely to have CS birth compared housewife women (AOR = 2.28(95%CI1.39,3.75)) and women self-employed was 3.73 times more likely to have CS birth compared with housewife women.(AOR = 3.73(95% CI1.62,8.59)), a prim para women was 6.79 times more likely to have caesarean birth compared with grand multi para women (AOR = 6.79(95%CI2.02,22.79)) and para two women were 3.88 times more likely to have caesarean birth (AOR = 3.88(95% CI1.15,13.08)) than grand multi para women. Another factor is the wealth index of the family, being the highest wealth index was 5.39 times more likely to have caesarean birth than the lowest wealth index (AOR = 5.39 (95%CI 1.08, 26.8)) (**S2 File**).

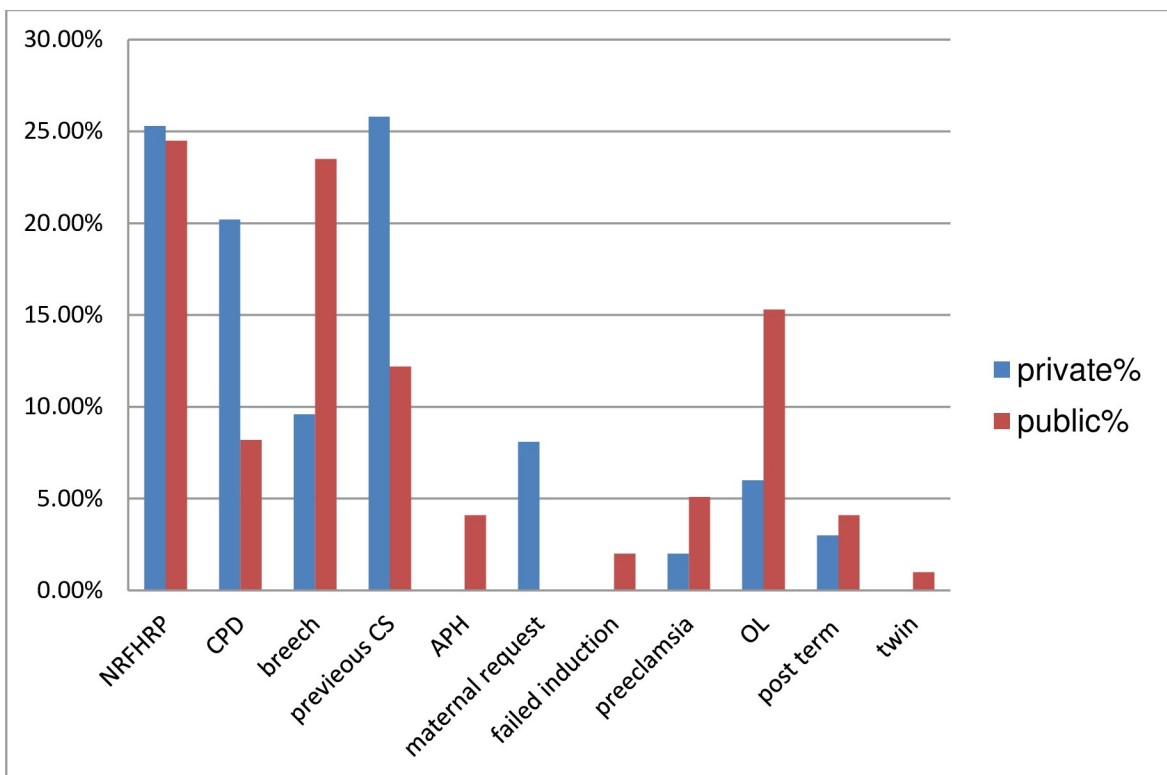

**Fig 2. Indications of caesarean birth of women who delivered in the selected public and private health facility in Bahir Dar city, Amhara Regional State, Ethiopia 2019.**

In this study, from the overall multivariable logistic regression model variables statistically associated with overall caesarean birth were: women who delivered at private health facility were 3.45 times more likely to have caesarean birth compared public health facility (AOR = 3.45(95%CI2.24,5.34)),women who have higher education level were 2.64 times more likely to have caesarean birth compared with women had no formal education (AOR = 2.64 (95%CI1.31,5.3)), being urban resident were 4.8 times more likely to have caesarean birth than rural (AOR = 4.8(95%CI1.8,12.76)), being fetal breech presentation were 3.16 times more likely to have caesarean birth compared cephalic presentation(AOR = 3.16(95%CI1.56,6.39)) and referred women were 2.71 times more likely to have caesarean birth compared its counterpart (AOR = 2.71(95%CI1.6,4.59))(**Table 3**).

## Discussion

This study findings revealed that significant higher caesarean birth rates in private health facility than public health facility delivery.

The results of the study revealed that the prevalence of caesarean birth in public health facility was 27.5% (95%CI: 22.8–32.2) which is consistent with other studies conducted in Attet Hospital, Gurage zone 27.6%[25], Gondar 27%[26],Felege Hiwote 25.4%[23], Addis Ababa 31.1%(14) and Harare 26.6%(20). But higher than studies conducted in Morocco 17.83% and India 13.7%[24, 27]. This might be due to that our study settings the selected health facilities serve as the main referral centers for most complicated pregnancies in the city and around the city.

**Table 3. Factors associated with caesarean birth among women who gave birth in selected public and private health facilities in Bahir Dar city, Amhara Regional State, Ethiopia, 2019 (n = 708).**

| Factors | Category | Caesarean birth | | COR (95%CI) | AOR (95%CI) |
|---|---|---|---|---|---|
| | | No | Yes | | |
| **Health facility type** | Public | 258 | 98 | 1 | 1 |
| | Private | 154 | 198 | **3.38(2.47,4.63)**\*\*\* | **3.45(2.24,5.34)**\*\*\* |
| **Age** | 15–24 | 117 | 47 | 0.47(0.28,0.79) | |
| | 25–34 | 242 | 204 | 0.99(0.64,1.54) | |
| | >35 | 53 | 45 | 1 | 1 |
| **Women Educational status** | No formal education | 86 | 24 | 1 | |
| | Primary | 88 | 40 | 1.62(0.90,2.92) | |
| | Secondary | 119 | 68 | **2.04(1.19,3.52)**\*\* | |
| | College and above | 119 | 164 | **4.93(2.96,8.22)**\*\* | **2.64(1.31,5.3)**\*\* |
| **Women occupation** | House wife | 267 | 133 | 1 | |
| | Government employee | 85 | 117 | **2.76(1.95,3.91)**\*\*\* | |
| | Self-employee | 59 | 45 | 1.53(0.98,2.37) | |
| | NGO | 1 | 1 | 2.0(0.12,32.34) | |
| **Spouse educational status** | No formal education | 86 | 16 | 1 | |
| | Primary | 45 | 20 | **2.38(1.12,5.05)**\* | |
| | Secondary | 85 | 57 | **3.60(1.91,6.77)**\*\*\* | |
| | College and above | 192 | 196 | **5.48(3.10,9.69)**\*\*\* | |
| **Residency** | Rural | 55 | 7 | 1 | 1 |
| | Urban | 357 | 289 | 6.36(2.85,14.17)\*\*\* | **4.8(1.8,12.76))**\*\*\* |
| **Gravida** | One | 156 | 179 | **2.49(1.44,4.27)**\*\* | |
| | Two | 83 | 108 | **2.19(1.23,3.89)**\*\* | |
| | Three | 36 | 65 | 1.58(0.83,3.0) | |
| | Four &above | 21 | 60 | 1 | |
| **Parity** | Para one | 162 | 181 | **2.55(1.49,4.38)**\*\* | |
| | Two | 80 | 106 | **2.15(1.21,3.83)**\*\* | |
| | Three | 33 | 65 | 1.45(0.75,2.77) | |
| | Four & above | 21 | 60 | | |
| **Gestational age(weeks)** | <37 | 25 | 9 | 1 | |
| | 37–42 | 384 | 280 | 2.02(0.93,4.4) | |
| | >42 | 3 | 7 | **6.48(1.37,30.6)**\* | |
| **Fetal presentation** | Cephalic | 390 | 267 | 1 | 1 |
| | Breech | 22 | 29 | **1.92(1.08,3.42)**\*\*\* | **3.16(1.56,6.39)**\*\* |
| **Wealth index** | Lowest | 61 | 23 | 1 | |
| | Second | 106 | 73 | **1.82(1.03,3.2)**\* | |
| | Medium | 78 | 55 | **1.87(1.03,3.37)**\* | |
| | Fourth | 117 | 89 | **2.01(1.16,3.5)**\* | |
| | Highest | 49 | 56 | **3.03(1.64,5.61)**\*\*\* | |
| **Referral status** | Not referred | 307 | 242 | **1** | 1 |
| | Referred | 105 | 54 | **0.65(0.45,0.94)** | **2.71(1.60,4.59)**\*\*\* |

\*statically significant at 0.05 < P<0.01

\*\* 0.01<p<0.001

\*\*\*<0.001

The prevalence of caesarean birth in private health facilities was 56.3% (95%CI: 50.9–61.4), which is consistent with the study conducted in Harar 58.7%[20]. But higher than the study conducted in Addis Abeba 48.3%[14]. This could be due to increased access and utilization of the emergency obstetrics service with time deference. However, the current finding is lower than studies conducted in Brazil 87.9% and Mexico 85.6%[16, 24]. This difference might be explained by a difference in accesses to the service, infrastructure, and socio-economic differences between countries.

The difference of caesarean birth among public and private health facilities was significantly different (P<0.001), and how that caesarean birth much more common in private health facilities. This finding was supported by other studies conducted in a different settings [20, 21, 28] and Addis Ababa[14]. This could be due to that in private health facility women delivered by cesarean section with women request indication. Moreover, private facilities are business-oriented and the procedure might be done without clear medical indication.

Type of health facility, the women who delivered in private health facilities were 3.45 times more likely to have caesarean birth as compared to women delivered in public health facilities. This finding is supported by studies conducted in other settings in Ethiopia [20, 21] and another country [15]. The possible explanation might be higher private health facility profits and higher provider remuneration for a CS delivery, provider's convenience of CS procedure and relatively lesser time required per birth than spontaneous vaginal delivery/assisted instrumental vaginal delivery.

The overall prevalence of caesarean birth in this study was 296 (41.8%) (95% CI: 38.4, 45.5). This finding is higher than studies conducted in Addis Abeba 38.3% and Harar 34.3%[17, 20]. This difference might be explained by increased access to the intervention, as it is observed by a large number of health facilities started providing the caesarean birth services in the study area and urban residents of the study area needs to be in to account. The prevalence of caesarean birth in this study area far exceeds with the WHO recommended a maximum limit of 15% cesarean section for any geographic area [29]. The reason could be due to that, WHO recommended among all delivery whereas in our country most of the delivery happened at home, as a result, the high prevalence might be attributed due only pregnant women perceived the risk of childbirth give birth at the health facility.

From public health facility women who have a breech presentation of the fetus were 3.64 times more likely to have caesarean birth compared with cephalic presentation This finding is consistent with similar studies done in India and Felge Hiwote hospital [23, 24]. This might be due to breech presentation is considered as one of the clinical indications by most of the providers to caesarean birth for the benefit of the fetus as well as the mother. The women who were referred fromother health facilities were 2.44 times more likely to have caesarean birth compared to self-referred women). This finding is supported by other previous study [23]. This might be due to the majority of referred women from other health facilities might have had some obstetric complications including complications requiring operative intervention. Residency, women who were from urban resident was 6.54 times more likely to have caesarean birth compared with rural dweller. This finding is supported by similar studies done in other settings [10, 16, 24]. The possible reasons for this could be most rural women cannot afford to deliver in urban private health facilities and usually delivered in rural public settings that have a limited capacity to provide CS, unless they are referred to higher public health facility to the study area and a role of the private health facilities in providing CS to wealthier women mostly in the study area.

In the multivariable logistic regression factors significantly associated with caesarean birth in private health facilities were women aged between 15–24 years was 80% less likely to have caesarean birth as compared to women age 35 and above. This finding was supported by other

studies done in Addis Ababa, Mexico, and India[16, 17, 24]. The possible explanation might medical conditions that led to caesarean birth like hypertension, diabetes, and macrosomia being more prevalent at an older age group. The odds of the women being governmental employees 2.72 times more likely to have caesarean birth and self-employed women were 3.1 times more likely to have caesarean birth compared to housewife women. This finding might be explained by the financial capacity of these women who could afford the fee of CS service provided by private health facilities and some of these women also had the privilege of health insurance from their working organization for covering the cost of the service. Para one woman was 6.79 times more likely and para two women were 3.88 times more likely to have caesarean birth as compared to grand multipara women. This finding was supported by studies done in Addis Ababa Ethiopia and other countries [16, 17, 24]. The possible explanation might be to avoid the arduous process of labor and delivery for para one mothers and previous mode of delivery for para two mothers, who have had a traumatic previous birth or complications, or believe incorrectly that vaginal birth is not possible after a previous CS. This finding is a matter of concern particularly for para one mothers, since this contributes to increased further caesarean birth, because of the previous history of caesarean birth is one of the critical indications for CS in subsequent delivery. The other significantly associated factor was the wealth index of the family, women with the highest-level wealth asset were 5.39 times more likely to have caesarean birth as compared to lowest level wealth asset. This result is supported by previous studies done in Harer and Mexico[16, 20]. This finding elucidates that caesarean birth seems to be a choice method for a woman who can afford it rather than being a procedure for safe delivery when medically indicated. Understandably, women with the highest wealth index status prefer to attend in the private health facilities to avoid all the administrative procedure in the public health facilities that are also associated with poor medical attention due to the larger quantity of women that each provider has to attend daily.

Women who had college diplomas and above were 2.64 times more likely to have caesarean birth as compared to women with no formal education. This finding supported by studies conducted in Addis Ababa and Mexico[16, 17]. The reason might be due to the high confidence of most educated women on modern medicine like the effectiveness of CS delivery and considering caesarean birth have less painful, convenient for selecting their delivery date and safer option than vaginal birth.

## Limitation of the study

The study used a quantitative approach alone to collect the data; triangulation with qualitative approach may have been more useful in addressing provider-related factors, for instance, the study did not evaluate the institutional/obstetrician factors such as performing cesarean section for teaching the purpose, economic incentives, time management, and medico-legal issue risk-minimizing behavior.

## Conclusions

The prevalence of caesarean birth in both public and private health facility found to be high. In this study, the prevalence of cesarean section delivery in private health facilities was more than twice as high as that of a public health facility. The prevalence of caesarean birth in public and private health facilities has statistically significant difference.

The breech presentation compared with the cephalic presentation, women referred compared with not referred women and urban residences compared with its counter were variables significantly associated with caesarean birth in a public health facility. Whereas, the age of mother 15–24 years, governmental and self-employed women than housewife women, Para

one and two women compared with grand multi para women and being highest level of wealth asset than lowest were variables significantly associated with caesarean birth among private health facility.

In the full model public facility compared with the private health facility, women who have a college diploma and above compared with no formal education, urban residence compared with rural, breech presentation compared with cephalic and referred compared with not referred women were variables significantly associated with caesarean birth in overall health facilities in the study area.

## Recommendations

Tailored information and support about childbirth fear, pain relief, and indication of caesarean sections birth is required focusing urban dweller women. Health care provider is necessary to advocate vaginal delivery and women shall be fully informed about the risks associated with medically unjustified cesarean section in private health facility. Medical audit of labor management both in private and public health facilities is warranted to maintained the caesarean birth rate with the level of WHO recommendation. Further triangulated study to explore provider related factors and to fully understand why higher educational level women and those women who afforded the fee of private health facility preferred caesarean birth.

## Supporting information

**S1 File. English and Amharic version questionnaires.**
(DOCX)

**S2 File. Factors associated with caesarean birth among women who gave birth in public (n = 356) and private (n = 352) health facilities in Bahir Dar city, Amhara Regional State, Ethiopia, 2019.**
(DOCX)

**S1 Table.**
(DOCX)

## Acknowledgments

We would like to forward my deepest appreciation and thanks to the University of Gondar, College of medicine and health sciences, and the Institute of public health for providing Ethical clearance. We are deeply grateful to study participants, data collectors, and supervisors who participated in this study. We would also like to extend our thanks to the Health facility administrator for the provision of support letter and staff cooperation during the data collection.

## Author Contributions

**Conceptualization:** Meseret Bantigegn Melesse, Alehegn Bishaw Geremew, Solomon Mekonnen Abebe.

**Data curation:** Meseret Bantigegn Melesse, Alehegn Bishaw Geremew.

**Formal analysis:** Meseret Bantigegn Melesse, Alehegn Bishaw Geremew, Solomon Mekonnen Abebe.

**Investigation:** Meseret Bantigegn Melesse.

**Methodology:** Meseret Bantigegn Melesse, Alehegn Bishaw Geremew, Solomon Mekonnen Abebe.

**Project administration:** Meseret Bantigegn Melesse.

**Resources:** Meseret Bantigegn Melesse.

**Software:** Meseret Bantigegn Melesse, Alehegn Bishaw Geremew.

**Supervision:** Meseret Bantigegn Melesse, Alehegn Bishaw Geremew, Solomon Mekonnen Abebe.

**Validation:** Meseret Bantigegn Melesse, Alehegn Bishaw Geremew.

**Visualization:** Meseret Bantigegn Melesse.

**Writing – original draft:** Meseret Bantigegn Melesse, Alehegn Bishaw Geremew.

**Writing – review & editing:** Alehegn Bishaw Geremew, Solomon Mekonnen Abebe.

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
