## [Decision Letter · Decision Letter 0]

4 Feb 2020

PONE-D-19-32522

High prevalence of cesarean section delivery among health facilities delivered mothers in Bahir Dar city, Amhara region, Ethiopia.   a comparative study

PLOS ONE

Dear Mr. Geremew,

Thank you for submitting your manuscript to PLOS ONE. After careful consideration, we feel that it has merit but does not fully meet PLOS ONE’s publication criteria as it currently stands. Therefore, we invite you to submit a revised version of the manuscript that addresses the points raised during the review process.

This manuscript has major grammatical errors and overall not written well. The objective is unclear from the beginning. The methods need to be fleshed out more. The results are hard to follow. All sections of the manuscript need to be overhauled significantly. Please make an extensive English revision, by a native English speaking person. And besides this, there are specific comments from Reviewers.

We would appreciate receiving your revised manuscript by Mar 20 2020 11:59PM. To enhance the reproducibility of your results, we recommend that if applicable you deposit your laboratory protocols in protocols.io, where a protocol can be assigned its own identifier (DOI) such that it can be cited independently in the future. For instructions see: http://journals.plos.org/plosone/s/submission-guidelines#loc-laboratory-protocols

We look forward to receiving your revised manuscript.

Kind regards,

Ricardo Q. Gurgel, PhD

Academic Editor

PLOS ONE

Journal Requirements:

2. Please include additional information regarding the survey or questionnaire used in the study and ensure that you have provided sufficient details that others could replicate the analyses. For instance, if you developed a questionnaire as part of this study and it is not under a copyright more restrictive than CC-BY, please include a copy, in both the original language and English, as Supporting Information.  If the original language is written in non-Latin characters, for example Amharic, Chinese, or Korean, please use a file format that ensures these characters are visible.

3. Please state whether you validated the questionnaire prior to testing on study participants. Please provide details regarding the validation group within the methods section.

4. Thank you for stating the following in the Funding Section of your manuscript:

"The source of fund for this study was Amhara regional health bureau."

Reviewers' comments:

Reviewer's Responses to Questions

**Comments to the Author**

1. Is the manuscript technically sound, and do the data support the conclusions?

Reviewer #1: Yes

Reviewer #2: Partly

2. Has the statistical analysis been performed appropriately and rigorously? 

Reviewer #1: Yes

Reviewer #2: I Don't Know

3. Have the authors made all data underlying the findings in their manuscript fully available?

Reviewer #1: Yes

Reviewer #2: No

4. Is the manuscript presented in an intelligible fashion and written in standard English?

Reviewer #1: Yes

Reviewer #2: No

5. Review Comments to the Author

Reviewer #1: - preterm gestational age in private facilities table 2; n = 3/352 does not equal 85.2%

- recommend using term "cesarean birth" rather than "cesarean delivery"

- paper is limited by lack of outcomes data; we do not know how higher cesarean birth rate translates to maternal and perinatal outcomes

Reviewer #2: - Line 24 of Abstract correct spelling of “assess”

- Line 24 spell out acronym of CS

- Line 25 grammatical error “public and private health facilities delivered mother…” difficult to understand main objective of manuscript

- Line 28 indicate number of health facilities

- Line 50 in definition of CS, add “of the mother”

- Line 50 add the word “mother” after “when adequately indicated”

- Line 52 specify if globally

- Line 53 no comma after “But”

- Line 58 replace “cesarean section” with CS. Check acronym use throughout manuscript

- Line 60 needs citation

- Line 64 specify if global

- Line 73 specify context – is this in Ethiopia or Sub-Saharan Africa?

- Line 75 remove extra period before sentence

- Line 85 make setting plural to settings

- Line 89 clarify this sentence. What are the “both situations are unwanted” referring to?

- Line 103 How many health facilities? Specify public and private health facilities. State earlier.

- Line 107 indicate percentage of female population rather than number

- Line 107 – 112 provide an overview of types of health facilities but which provide CS?

- Line 120 indicate age group of women and is it in all public and private health facilities in Bahir Dar?

- Sample Size please clarify methodology and tighten

- Line 135 correct first sentence and add the word “was” to read “was used” rather than “use used.”

- Line 137 correct spelling of Bahir Dar

- Line 143 suggest table with health facilities and number of participants

- Operational Definitions include within paragraph rather than separate sub-heading

- Line 159 what is the name of the questionnaire and checklist? What is the difference between the questionnaire and the checklist?

- Line 170 specify how much time after delivery women were interviewed. Indicate range of how many hours.

- Line 202 mention informed consent of participants earlier perhaps around line 159

- Line 202-203 include IRB number

- Line 207 verbal consent indicate if this is due to literacy rates

- Discussion: Comparison to CS prevalance in other countries seems out of context, some data in discussion seem to be better fit in Results section, consider reducing Discussion a lot

- Line 325 why CS more prevalent in private health facilities versus public – perhaps due to economical reasons? Why would these women request CS more? More reproductive autonomy than women who deliver in public facilities? Why?

o This is discussed more at the end of the Discussion, but consider weaving in sooner.

- Recommendations seem out of place with manuscript

- Figure – label Y axis

General comments:

- Grammatical errors throughout make the manuscript difficult to follow (spelling, capitalization, punctuation, etc.)

- Consistent use of acronyms throughout manuscript

- Methods need to be fleshed out more

- Results hard to follow, reformat tables for easier readability and prioritize variables

6. PLOS authors have the option to publish the peer review history of their article (what does this mean?). If published, this will include your full peer review and any attached files.

Reviewer #1: Yes: Margo S Harrison

Reviewer #2: No

---

## [Author Response · Author response to Decision Letter 0]

3 Mar 2020

Authors' point-by-point response to reviewers reports to manuscript code PONE-D-19-32522

Title: - High prevalence of caesarean section delivery among health facilities delivered mothers in Bahir Dar city, Amhara region, Ethiopia. a comparative study 

First of all, the authors would like to thank PLose one Editor and reviewers providing the necessary comments and suggestion which are very crucial to improve our manuscript. The authors have made corrections point by point to the comments given and questions raised by editor and reviewers. Please note that we gave our response in blue font colour for editor and reviewers comments. 

Response to editor 

Journal Requirements:

 Response 

The suggestion has accepted and we have tried to adhere the journal requirement 

2. Please include additional information regarding the survey or questionnaire used in the study and ensure that you have provided sufficient details that others could replicate the analyses. For instance, if you developed a questionnaire as part of this study and it is not under a copyright more restrictive than CC-BY, please include a copy, in both the original language and English, as Supporting Information. If the original language is written in non-Latin characters, for example Amharic, Chinese, or Korean, please use a file format that ensures these characters are visible.

Response 

We have included the data collection tool as additional file 1, we have prepared the English version questionnaires and checklist, and for data collection we have used translated Amharic vesrrion questionnaires and English version checklist because of the check list was used to extract data from the chart 

3. Please state whether you validated the questionnaire prior to testing on study participants. Please provide details regarding the validation group within the methods section.

4. Thank you for stating the following in the Funding Section of your manuscript:

"The source of fund for this study was Amhara regional health bureau."

"The funders had no role in study design, data collection, decision to publish, or preparation of the manuscript."

Response

Thank you for comments we have corrected as follow: The funder had no role in decision to publish or preparation of the manuscript

Response 

We have linked with ORICD ID

Reviewers' comments:

Reviewer's Responses to Questions

Comments to the Author

1. Is the manuscript technically sound, and do the data support the conclusions?

Reviewer #1: Yes

Reviewer #2: Partly

2. Has the statistical analysis been performed appropriately and rigorously? 

Reviewer #1: Yes

Reviewer #2: I Don't Know

3. Have the authors made all data underlying the findings in their manuscript fully available?

Reviewer #1: Yes

Reviewer #2: No

4. Is the manuscript presented in an intelligible fashion and written in standard English?

Reviewer #1: Yes

Reviewer #2: No

5. Review Comments to the Author

Response to reviewer 1

Reviewer #1: - preterm gestational age in private facilities table 2; n = 3/352 does not equal 85.2%

Response

We admired reviewer for the comment has given, it was type error and has corrected to 0.85% instead of 85.2%.

- recommend using term "cesarean birth" rather than "cesarean delivery"

Response

- paper is limited by lack of outcomes data; we do not know how higher cesarean birth rate translates to maternal and perinatal outcomes

Response

You are correct in our results part there is no maternal and perinatal outcome because our objectives was to ass caesarean birth rate and compare it among private and public facility. however, as a known evidence we have stated the impact of unnecessary CS on maternal outcome. We have collected the outcome data but by considering that is not the study objective we did not included. If you request more to add it is possible to state the outcomes data under descriptive part in the next recommendation 

Response to reviewer 1

Reviewer #2: - Line 24 of Abstract correct spelling of “assess”

Response

Thank you, we have corrected it 

- Line 24 spell out acronym of CS

Response

 Thank you we have accepted and according to comment given by reviewer 1 totally we have changed caesarean delivery by caesarean birth

- Line 25 grammatical error “public and private health facilities delivered mother…” difficult to understand main objective of manuscript

Response

Greatly we thank you, we have corrected as women gave birth at public and private health facilities

- Line 28 indicate number of health facilities

Response

Accepted and has indicated 

- Line 50 in definition of CS, add “of the mother”

Response

We have corrected 

- Line 50 add the word “mother” after “when adequately indicated”

Response

We authors agreed the word mother is not necessary after when adequately indicated because the indication of caesarean delivery is not limited to mother request. The indication status can be determined by clinician and it can be fetal problem, labour condition or maternal condition. If you convince us next time, we can consider correction.

- Line 52 specify if globally

Response

Corrected 

- Line 53 no comma after “But”

Response

Corrected

- Line 58 replace “cesarean section” with CS. Check acronym use throughout manuscript

Response

We have tried to use the word caesarean birth 

- Line 60 needs citation

We revised the sentence and the citation is the same with the next sentence 

Response

- Line 64 specify if global

Response

Corrected

- Line 73 specify context – is this in Ethiopia or Sub-Saharan Africa?

Response

We have accepted the comment and corrected as in Ethiopia

- Line 75 remove extra period before sentence

Response

Corrected 

- Line 85 make setting plural to settings

Response 

Corrected 

- Line 89 clarify this sentence. What are the “both situations are unwanted” referring to?

Response

Context where insufficient caesarean birth rate (<5%) and inappropriate high caesarean birth rate (>15%)

- Line 103 How many health facilities? Specify public and private health facilities. State earlier.

Response

 A total of 9(4 public and 5 private) health facility provide caesarean section birth in Bahir Dar city during the study period and 3 facility from public and 4 facility from private facility were selected and we have used design effect since we did not collect data from the 9 facility. 

- Line 107 indicate percentage of female population rather than number

Response

We have accepted and consider both number and percentage 

- Line 107 – 112 provide an overview of types of health facilities but which provide CS?

Response

Among all health facilities 4 public facilities named Tibebe Giwon, Felege Hiwot,Adiss Alem, Bahir Dar health center, and 5 private health facilities named GAMBY hospital, Mari stop, Addinas General hospital, and Dr. Amiro MCH specialty clinic were provide CS. We have stated briefly from the sampling procedure.

- Line 120 indicate age group of women and is it in all public and private health facilities in Bahir Dar?

Response 

Yes, the figure is from all public and private facility which provide delivery service 

- Sample Size please clarify methodology and tighten

Response

Thank you, we have accepted the comment we did not include in the manuscript from the previous submission because of the manuscript has been long we have clarified it from the tack change manuscript on page number 7-8 and also as has described follow:

N (in each group) = (p1q1 + p2q2) (f(�,�)) / ((p1 - p2)²

Where n= sample size for each group 

P1= the proportion of CS at Private health facilities taken from Addis Ababa 2017(0.47). 

P2= the proportion of CS at Public health facilities taken from Addis Ababa 2017(0.34).

F (�,�) =7.84, when the power = 80% and the level of significance = 5%

q1= (1-p1) =1-0.47=0.53

q2= (1-p2) =1-0.34=0.66

n1=n2 = (0.84+1.96)2 ((.47�.53) + (.34 �.66)) / (.47 - .34)2 = 219

Total sample size for both groups =438

To address the study design effect the total sample size is multiplied by1.5

I.e. 438 *1.5 =657 

 Adding 10% none response rate, the final estimated total sample size for the study was 724

- Line 135 correct first sentence and add the word “was” to read “was used” rather than “use used.”

Response

Sorry, we have corrected 

- Line 137 correct spelling of Bahir Dar

Response

Sorry, we have corrected 

- Line 143 suggest table with health facilities and number of participants

Response

 Respected reviewer, we did the schematic presentation of sampling procedure while we did our research work but we did not submit in the manuscript due to our perceptions of not necessary and it can make the manuscript too long. The procedure is the following we can consider if you recommended to add in the manuscript.

- Operational Definitions include within paragraph rather than separate sub-heading

Response

We have accepted and corrected 

- Line 159 what is the name of the questionnaire and checklist? What is the difference between the questionnaire and the checklist?

Response

Structured interview-based questionnaire and data abstraction checklist. The difference between the two is interview based questionnaire was used to collect primary data from the participants and data abstraction checklist was used to collected data from maternal history chart and delivery summary.

- Line 170 specify how much time after delivery women were interviewed. Indicate range of how many hours.

Response

Accepted and has corrected 

- Line 202 mention informed consent of participants earlier perhaps around line 159

Response

We have already stated under line 223

- Line 202-203 include IRB number 

Response

Accepted and we have added the IRB number from track change manuscript on page number 12 IPH/180/06/2022

- Line 207 verbal consent indicates if this is due to literacy rates

Response

The reason why we secured verbal informed consent rather written was not due to literacy rate. Of course, study include women cannot read and write. The mean reason why verbal consent was we did not too biological sample or give/denied any treatment for participants for the purpose of this research as a result we secured only informed verbal consent. 

- Discussion: Comparison to CS prevalance in other countries seems out of context, some data in discussion seem to be better fit in Results section, consider reducing Discussion a lot

Response

Of course, considering context during discussion of our finding is necessary, however for this particular research outcome t WHO recommended the caesarean birth rate to be 5-15% if at health facility and not more than 10 percent at population level for any context. This is why we have discussed our finding with any settings.

Thank you, we have accepted and deleted some points seems like result what we already reported from the result section of the track change version. We have tried to reduce the discussion by removing word/sentence that may not affect the discussion. However, we authors agreed and did not delete discussion form variables associated with caesarean birth. If you recommended/suggest to delete in the next time we can admit and correct it

- Line 325 why CS more prevalent in private health facilities versus public – perhaps due to economical reasons? Why would these women request CS more? More reproductive autonomy than women who deliver in public facilities? Why?

o This is discussed more at the end of the Discussion, but consider weaving in sooner.

Response

There a number of reasons for high CS in private compared Public health facility as we have discussed, women have health insurance by the employee sector attend private facility and there was CS done due to women request in private but not observed in public facility. Most women have reproductive and other autonomy visit private facility. Most educated and urban dweller women attend private facility. Other that we did not addressed in our study private organization established for profit as a result unnecessary CS might be high in private.

In general, in our study, there are factors specifically from private facility associated with caesarean birth such as women from highest wealth index, occupation government employed.

The sequence of the discussion is written according to the results

- Recommendations seem out of place with manuscript

Response

We authors thank you for your critical comments. We have corrected from line 492-500 on the track change document 

- Figure – label Y axis

Response

We have corrected it

General comments:

- Grammatical errors throughout make the manuscript difficult to follow (spelling, capitalization, punctuation, etc.)

- Consistent use of acronyms throughout manuscript

- Methods need to be fleshed out more

- Results hard to follow, reformat tables for easier readability and prioritize variables

Response 

We have accepted the comments and suggestion given. We have considered general comments has given to improve the manuscript. We have deleted one whole table and has reported the result only text, and also one table described by graph. 

6. PLOS authors have the option to publish the peer review history of their article (what does this mean?). If published, this will include your full peer review and any attached files.

Do you want your identity to be public for this peer review? For information about this choice, including consent withdrawal, please see our Privacy Policy.

Reviewer #1: Yes: Margo S Harrison

Reviewer #2: No

---

## [Editor Report · Decision Letter 1]

30 Mar 2020

High prevalence of caesarean birth among mothers delivered at health facilities in Bahir Dar city, Amhara region, Ethiopia.   a comparative study

PONE-D-19-32522R1

Dear Dr. Geremew,

We are pleased to inform you that your manuscript has been judged scientifically suitable for publication and will be formally accepted for publication once it complies with all outstanding technical requirements.

With kind regards,

Ricardo Q. Gurgel, PhD

Academic Editor

PLOS ONE
---

## [Editor Report · Acceptance letter]

3 Apr 2020

PONE-D-19-32522R1 

High prevalence of caesarean birth among mothers delivered at health facilities in Bahir Dar city, Amhara region, Ethiopia.   a comparative study 

Dear Dr. Geremew:

I am pleased to inform you that your manuscript has been deemed suitable for publication in PLOS ONE. Congratulations! Your manuscript is now with our production department. 

With kind regards,

on behalf of

Professor Ricardo Q. Gurgel 

Academic Editor

PLOS ONE